# Ultrafast dynamic stark shift of an exciton-polariton condensate

Sarit Feldman [1] ✉, Dmitry Panna [1], Nadav Landau[1], Sebastian Brodbeck[2], Sebastian Klembt [2], Christian Schneider [2,3], Sven Höfling[2] & Alex Hayat[1]

Time-dependent spatial modulation of a quantum condensate plays a key role in the investigation of the dynamics of collective coherent systems. Cold-atom condensates were already manipulated into forming solitons, vortices and optical lattices via the dynamic Stark effect. In the solid-state framework, semiconductor microcavity exciton-polaritons present an excellent platform for observing nonequilibrium condensate dynamics. Furthermore, the dynamic manipulation of exciton-polariton condensates offers numerous potential implementations in all-optical logic devices. We report the first observation of the energy modulation of an exciton-polariton condensate via the dynamic Stark effect. A Stark pulse is employed to induce a transient, coherence-preserving blueshift in the energy of the condensate of femtosecond-scale duration. A novel approach is presented for discerning the Stark effect in a condensate from the uncondensed case according to distinctive features in differential reflectance spectra, including spectro-temporal oscillations enabling to detect the buildup of coherence. The ultrafast and non-invasive modulation of the exciton-polariton condensate demonstrated here opens new research avenues of nonequilibrium physics of solid-state condensates as well as opportunities for polariton-based elements in both classical and quantum information technologies.

The strong coupling between the cavity vacuum mode and quantum well (QW) excitons in a semiconductor microcavity yields the formation of composite bosonic quasiparticles, the exciton-polaritons, combining light and matter properties. Exciton-polaritons present a platform for phenomena of both fundamental and practical interest, one of them being the transition to a macroscopic quantum condensed state[1] at high temperatures up to room temperature[2,3], exhibiting low-threshold lasing[4–6]. The spatial modulation of quantum condensates represents a stepping-stone to exploring physical aspects of quantum hydrodynamics and topological states. In the field of cold atoms, the spatial and dynamical manipulation of the profiles of Bose-Einstein condensates (BEC) led to the confinement of the BEC in optical lattices[7,8] and to the generation of solitons[9,10] and vortices[11]. To induce these phenomena, the dynamic Stark effect by an off-resonant

beam was employed, generating a coherence-preserving transient shift in the energy of the state. Exciton-polariton condensates provide a solid-state alternative to cold atoms enabling the observation of new states of the matter and gaining insights into nonequilibrium condensation dynamics in a scalable platform operating at accessible temperatures. Phase imprinting and potential shaping of polariton condensates have been shown to induce phenomena such as solitons, polariton superfluidity and quantized vortices[12–15]. The manipulation of polariton condensates holds promise for applications in classical and quantum technologies including all-optical circuits[16–21], quantum simulators[22,23] and quantum networks[24] The increasingly high propagation speeds[25,26], as well as the strong nonlinearity arising from efficient interactions between polaritons facilitates the achievement of high operational gains[21] and complex operations[27] in logic devices.

[1]Department of Electrical and Computer Engineering, Technion – Israel Institute of Technology, Haifa, Israel. [2]Technische Physik, Universität Würzburg, Am Hubland, Würzburg, Germany. [3]Institute of Physics, Carl von Ossietzky University, Oldenburg, Germany. ✉e-mail: saritf@campus.technion.ac.il

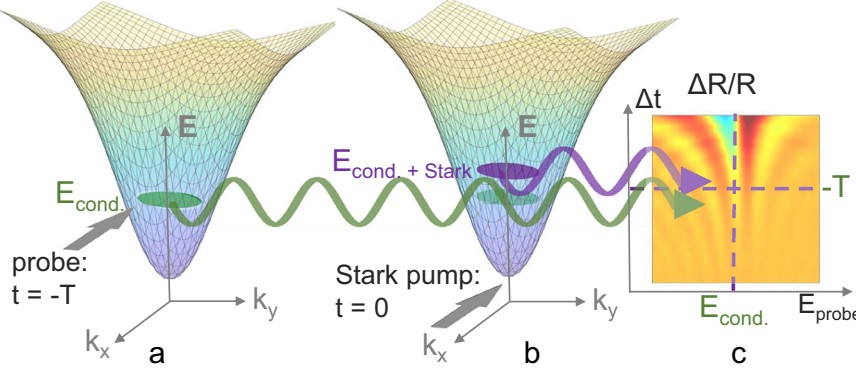

**Fig. 1 | Coherent oscillations in a condensate formed in the LP branch. a** Buildup of polarization in probe-induced LP condensate, starting from probe incidence at $t = -T$, and reemission (green arrow) prior to Stark beam incidence. **b** Incidence of the Stark beam at $t = 0$ modifies the condensate energy. Emission is depicted by the purple arrow. Interference of the polarization in (**a**) and (**b**) results in spectro-temporal oscillations in the differential reflectivity (**c**). $E_{cond.}$ and $E_{cond+Stark}$ represent the condensate energy in the absence and in the presence of the Stark beam, respectively.

Exciton-polariton condensates enable the formation of topologically protected states maintaining coherence over long distances[28], which is of interest for quantum computational schemes. Existing methods for dynamically shaping the potential of polariton condensates are usually realized through repulsive interactions with carriers injected by various pumping schemes[17,18,29–31] that irreversibly modify the condensate wavefunction and are often constrained to the duration of the carrier decay; while alternative, non-invasive modulation methods nevertheless operate at ns time scales[32].The resulting switching rates do not suffice for manipulating states within the condensate coherence time. It has been demonstrated in previous work that the ultrafast and reversible modulation of the polariton energy level structure can be achieved via the dynamic Stark effect[33–35]. Until now, however, the Stark shift of a polariton condensate has not been demonstrated.

Here, we report the first modulation of a polariton condensate by the dynamic Stark effect. We developed a novel approach based on time-resolved differential reflectance spectroscopy for observing the Stark effect alongside the onset of condensation, ordinarily determined from photoluminescence (PL) measurements[1,36,37]. Our results demonstrate a reversible transient shift on the femtosecond time scale in the energy of the lower polariton (LP) condensate, induced by a Stark sub-resonant pulsed beam. We first identify the signature of the Stark shift below condensation threshold as the known short-lived spectral blueshift of the LP and upper polariton (UP) absorption dips[33–35]. The density of the polaritons is then increased up to LP condensation threshold, at which point the signature of the dynamic Stark effect is markedly altered by the combination of increased photoluminescence, enhanced repulsion-induced energy blueshift of the LP and the buildup of coherence. The latter is expressed in a change in spectro-temporal oscillations of the reflectance which, according to the theory of coherent transients[38–40], decay at a rate determined by the coherence of the induced polarization. Analysis of the decay rate of the oscillations reveals a sharp enhancement in the temporal coherence of the polariton system at threshold density. This behaviour, comparable to polariton condensates observed in standard PL measurements, provides evidence that the dynamic Stark effect is induced on polaritons in the condensed state.

## Results

### Coherent oscillations in differential reflectivity
A pump-probe experimental setup was employed to obtain time-resolved differential reflectance spectra. Unlike the dynamic Stark shift occurring, roughly, when the probe and the Stark pump pulses overlap in time, the coherent oscillations are observed at pump-probe delay $\Delta t < 0$, i.e. when the pump reaches the sample after the probe. A simplified picture of the origin of the oscillations can be depicted as follows (Fig.1): first, the probe triggers the buildup of polarization in the material, initiating reemission. The Stark pump pulse subsequently reaches the sample and modifies the part of the polarization generated in its presence. Interference of the initial part of the emission (Fig.1a) with the part altered by the Stark beam (Fig.1b) results in time-dependent spectral oscillations (Fig.1c). The decay rate of the fringes trends inversely to the temporal coherence of the polarization; upon formation of a polariton condensate, the buildup of temporal coherence and the narrowing of the PL spectral linewidth[1] result in increased duration of the fringes.

Measurements of the normalized differential reflectivity were performed at varying pump-probe delays $\Delta t$. While the probe spectrally overlaps with both the LP and the UP branches, the Stark pump is significantly red detuned to avoid carrier excitation by one-photon absorption. In our pump-probe scheme, the probe beam is also used for injecting polaritons resonantly. Condensation was obtained by scanning the average probe intensity through a range of 13–214 mW cm$^{-2}$, covering threshold. The differential reflectivity $\Delta R/R$ is given by:

$$\frac{\Delta R}{R} = \frac{R_{p+pr} - R_p}{R_{pr} - R_{bg}} \quad (1)$$

with $\Delta R$ being the Stark-induced change in reflectivity and the subscripts p + pr, p, pr and bg standing for pump and probe, pump, probe and background only, respectively. The pump-probe time delay is scanned in a range starting at slightly positive delays around $\Delta t \approx 0$ at which the pump precedes the probe, to negative delays of several picoseconds. The dynamic Stark effect of femtosecond duration around $\Delta t = 0$ is associated with the spectro-temporal coherent oscillations of the differential reflectivity at $\Delta t < 0$ (Fig.1c). We detect the buildup of coherence accompanying condensation through the enhanced prolongation of the oscillations above threshold density. A simulation of the differential reflectivity was implemented based on the theoretical model[38–40] (see Supplementary Note 1). The differential reflectivity was calculated according to the absorption in the presence of the Stark pump, given by:

$$\alpha(w) = \mathrm{Im}\, \frac{\Omega_p}{2\varepsilon_0 c} <\chi(\omega)> \quad (2)$$

and in the absence of the pump, assumed to be of the form:

$$\alpha_0(\omega) = \frac{\Omega_p}{2\varepsilon_0 c} \mathrm{Im}\left\{ \frac{i\mu^2}{\gamma + i(E_{pol}/\hbar - \omega)} \right\} \quad (3)$$

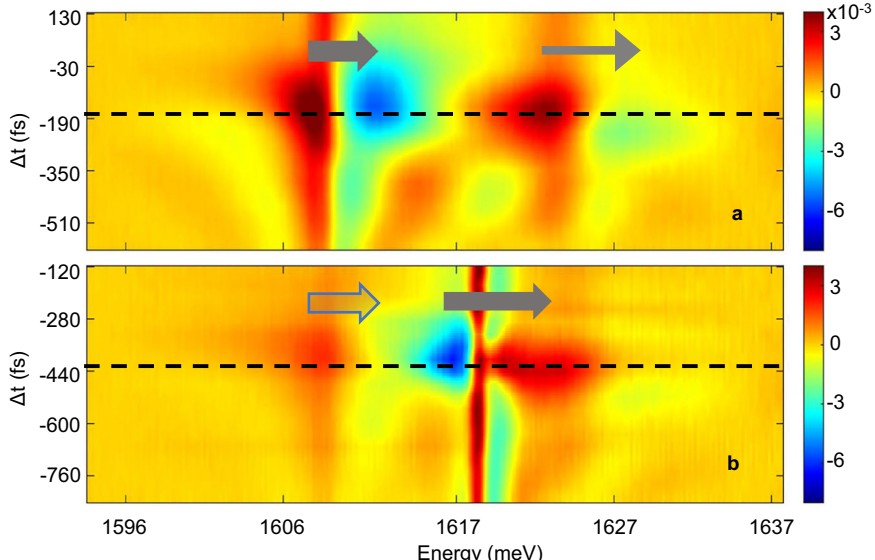

**Fig. 2 | Normalized differential reflectivity measured at high temporal resolution.** The measurements shown in (a),(b) were acquired around $\Delta t = 0$ at detuning $\delta = +5.1$ meV and at average probe intensity of 65 mW cm$^{-2}$ and 130 mW cm$^{-2}$, respectively, corresponding to a density below and above condensation threshold. a. Stark shift of the unpopulated branches of the LP (thick arrow) and the UP (thin arrow) branches. b. Stark effect acting on the co-existing LP states consisting of the uncondensed portion of the LP (thick bright arrow) and of the blueshifted polariton condensate, manifested as a shift of the PL energy (thick dark arrow). The dashed lines mark the pump-probe delay at which the effect is maximal.

where $\Omega_p$ is the pump frequency, $\mu$ is the polariton dipole moment, $\chi(\omega)$ is the material susceptibility, $E_{pol}$ is the polariton energy and $\gamma$ is the polariton decay rate. The linewidth of the polariton branches and the polariton energy as well as the pump-probe delay $\Delta t$ were extracted by fitting the calculated result for $\Delta R/R$ and the measurements.

## Stark effect on LP condensate

The differential reflectivity spectra measured at varying pump-probe delay $\Delta t$ and acquired at high temporal resolution around $\Delta t \approx 0$ are shown in Fig. 2 for a Stark beam peak intensity of 10.2 GW cm$^{-2}$ and at cavity-exciton detuning $\delta = +5.1$ meV, for which the LP is mainly excitonic. A Stark shift is exhibited in the polariton branches, as depicted by the arrows. Separate measurements conducted at an extended range of probe intensities and analysed in subsequent sections reveal the onset of a LP condensate at an average probe intensity of ~100 mW cm$^{-2}$. The measurements shown in Fig. 2a,b were conducted at average probe intensities of 65 mW cm$^{-2}$ and 130 mW cm$^{-2}$ in order to represent the cases of uncondensed and condensed LP, respectively.

Below condensation threshold (Fig. 2a), the dynamic Stark effect induces a blue shift of the absorption dips of the empty LP and UP polariton levels. Two clear distinctions from the uncondensed case emerge above threshold (Fig. 2b). The first is the blueshift in the energy of the LP branch, caused by polariton-polariton and polariton-exciton repulsive interactions and unrelated to the Stark shift induced by the pump. This blueshift, usually observed in direct PL measurements, is manifested here in the measured differential reflectivity. The second characteristic of condensation featured in Fig. 2b, is the modified signature of the dynamic Stark effect; unlike the uncondensed regime, a low energy dip is now observed alongside a higher energy peak. The explanation for this transition is in the fact that the Stark shift is now induced on highly populated states, therefore the PL becomes significant, and the absorption dip is replaced by a peak. The observed feature hence corresponds to the Stark shift of the PL of the condensate.

The pump-probe delay at which the Stark shift is largest was extracted from the fit to the model yielding, here, ~−190 fs for the uncondensed case and ~−440 fs for a condensate, as marked by dashed lines in Fig. 2b, respectively. Thus, in the condensed case the Stark pulse must be delayed further relative to the injection probe pulse to achieve the maximal effect. This result can be explained as follows. Below condensation threshold, the Stark effect appears as an energy shift of the reflectivity dips, with its timing corresponding to the buildup of the system polarization. Above the threshold, the Stark shift manifested in the condensate PL reaches its maximum once the condensate has formed. The delay of this shift corresponds to the time required for relaxation into the $k_\parallel$-0 states from higher energy states excited by the resonant, but broadband, probe beam. This contrast in delay times between the condensed and uncondensed systems persists across all probe intensities for $\delta = +5.1$ meV and $\delta \approx 0$ (Supplementary Fig. 3). The delay times for the condensate are slightly shorter for the exciton-like detuning of $\delta = +5.1$ meV compared to $\delta \approx 0$, consistent with the longer relaxation times expected for an increased excitonic fraction and a shallower dispersion curve.

Notably, the Stark effect is observed not only on the condensed, blueshifted LP but also on the low-energy LP that have not undergone condensation, as indicated in Fig. 2b by the light arrow for the uncondensed LP and by a dark arrow for the condensate. The coexistence of condensed and uncondensed LP spectral features arises from the nonuniform profile of the spot (the area populated by polaritons)[41]. The polariton density decreases towards the edges of the spot, leading to sub-threshold LPs at the periphery alongside with a condensate at the centre. Since the spectrometer collects the signal from a region larger than the spot, the reflectivity spectrum above threshold reflects the contributions of the uncondensed LPs at the edges, in addition to the condensate.

## Enhancement of coherence time at condensation

Figure 3 presents the normalized differential reflectivity at varying pump-probe delay $\Delta t$, sampled over an extended range of negative delays to trace the evolution of the coherent oscillations. The data shown are representative examples from a series of measurements performed at different probe (excitation) intensities. As in the previous section, the cavity−exciton detuning was fixed at $\delta = +5.1$ meV, while

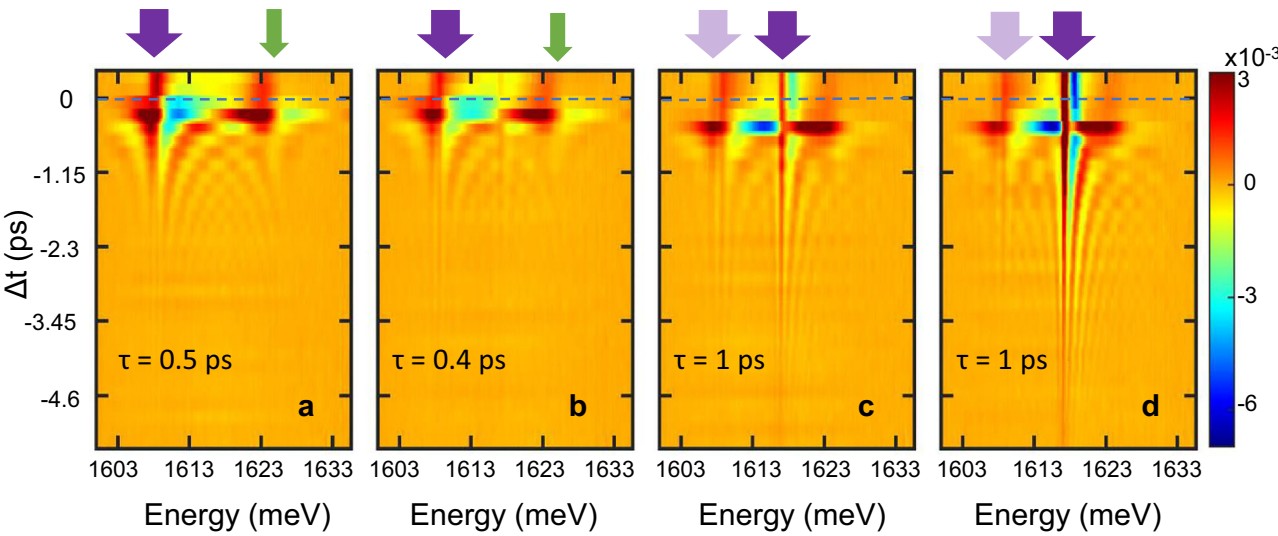

**Fig. 3 | Measured normalized differential reflectivity.** The probe injects polar-itons at an extended range of densities covering condensation threshold. Results in (**a**–**d**) are shown for an average probe intensity of 45 mW cm⁻², 92 mW cm⁻², 117 mW cm⁻² and 149 mW cm⁻², respectively at cavity-exciton detuning $\delta = +5.1$ meV and for a fixed pump peak intensity of 10.2 GW cm⁻². The LP and the UP are represented by thick purple arrows and green narrow arrows, respectively. The Stark shift at $\Delta t \approx 0$ is associated with the coherent oscillations at $\Delta t < 0$. The onset of condensation, observed in (**c**), is marked by a significant prolongation of the oscillations. Below condensation threshold (**a**, **b**), the oscillatory features are associated with the LP and UP. Above threshold (**c**, **d**), the energy of the major part of the LP polaritons is blueshifted, giving rise to the dominant, higher-energy oscillations, while the UP-associated features fade. The bright arrows in (**c**, **d**) represent the uncondensed portion of the LP. The dashed lines mark pump-probe temporal overlap ($\Delta t = 0$). $\tau$ represents the decay time constant of the oscillations.

the Stark beam was maintained at a peak intensity of 10.2 GW cm⁻². The results are displayed from left to right in order of increasing excitation intensity. The dynamic Stark shift of the LP and UP branches, observed at small delay times $|\Delta t|$, is accompanied by coherent oscillations at larger delay. The amplitude of the oscillations decays as $e^{-|\Delta t|/\tau}$ where the value of $\tau = 1/\gamma$, indicated for each measurement, is the polariton decay time. The measurements were also repeated for detuning $\delta \approx 0$ (shown in Supplementary Fig. 4).

Below a given probe-generated threshold polariton density, the decay time of the oscillations around the LP branch is inversely dependent on density (Fig. 3a, b). The increasing effect of repulsive polariton-polariton and polariton-exciton interactions[42] leads to decoherence and to the broadening of the PL linewidth. Above a threshold intensity, however, the trend is abruptly reversed, and the decay time increases with the density. The visibility of the fringes extends towards more negative time delay $\Delta t$ (Fig. 3c, d), indicating the spectral linewidth narrowing characterizing con-densation. The presence of prolonged oscillations unequivocally shows that the coherence of the condensate was preserved by the Stark beam. The extension of the temporal coherence above con-densation threshold is consistent with results reported with a similar excitation scheme comprising a resonant and normally incident beam[43], demonstrating the prolongation of the first order correlation $g^{(1)}$ of the PL.

An additional feature displayed in the condensed cases (Fig. 3c, d) consists in two adjacent spectral lines in the differential reflectivity of the condensed LP that are mostly present at $\Delta t > 0$, corresponding to times at which the Stark pump reaches the sample before the buildup of the probe-excited condensate. Contrarily to the Stark signature, these long lived and spectrally narrow lines indicate a redshift of the PL due to a long-term effect of the pump on the cavity via, e.g., a two-photon absorption (TPA)-induced change of the mir-ror refractive index[34]. The assumption of TPA is further supported by pump-induced PL measurements revealing an emission peak from the $Al_{0.2}Ga_{0.8}As$ layers of the top distributed Bragg reflector (DBR) and exhibiting a spectral shift attributed to carrier absorption (Supplementary Note 2).

## Characterization of LP linewidth and blueshift in differential reflectivity

The linewidth and the energy of the LP at varying injection intensity extracted from the fit to the coherent transient model, displaying good agreement with the experimental results, are shown in Fig. 4a–d for cavity-exciton detuning of $\delta = +5.1$ meV and $\delta = 0$. An example for the calculated results compared to experimental data at $\delta = +5.1$ meV is shown in Fig. 4e, f for varying pump-probe delay below and above condensation threshold, respectively.

The threshold-like character of condensation is manifested both in the linewidth and in the energy of the LP at positive cavity-exciton detuning ($\delta = +5.1$ meV). A pronounced narrowing of the extracted linewidth associated with the prolongation of the fringes is observed at threshold polariton density (Fig. 4a), accompanied by a sharp increase in the interaction-induced blueshift (Fig. 4b). At $\delta \approx 0$ the narrowing is observed as well, in which case, though, the blueshift is much smoother (Fig. 4c,d). A similar trend is observed in the angle resolved spectroscopy results reported by Panna et al.[34] for off-resonantly excited PL from the same sample at varying cavity-exciton detuning. As demonstrated in Fig. 4a, c, when the linewidth is not dominated by interaction-induced decoherence - namely at low probe intensities - it is narrower at $\delta \approx 0$ than at $\delta = +5.1$ meV, consistent with the larger photonic fraction of the LP at $\delta \approx 0$. Equivalently, the coherent oscil-lations decay more slowly. As the probe intensity is increased, resulting in higher polariton densities, the blueshift of the energy of the LP branch induced by the repulsive interactions grows. The high values of the blueshifted LP energy at cavity–exciton detuning $\delta = +5.1$ meV, compared to the energy values obtained at $\delta \approx 0$, characterize polar-iton condensates formed at positive detuning[45], where the excitonic fraction of the LP is enhanced. Measurements of the LP energy versus excitation intensity at $\delta \approx 0$ obtained using a PL setup, are provided for reference in Supplementary Note 3. To confirm that the blueshift ori-ginates from condensed polaritons and to rule out attributing it to photon lasing, we point out both to the short-lived signature of the Stark shift around $\Delta t \approx 0$, well above the bare exciton energy (Figs. 2b and 3c,d), and to the relaxation-induced prolongation above condensation threshold of the pump-probe delay at which the short-

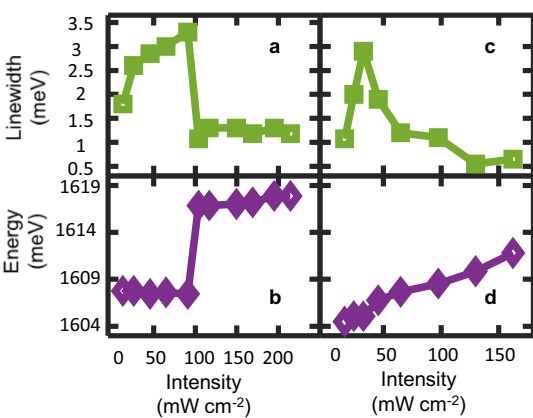

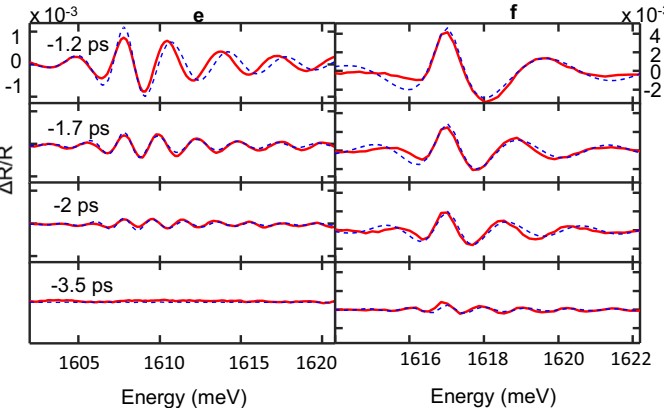

**Fig. 4 | Linewidth and energy of the LP branch vs. probe intensity. a, b** Linewidth (**a**) and energy (**b**) of the LP branch vs. average probe (injection) intensity at cavity-exciton detuning $\delta = +5.1$ meV. **c, d** Linewidth and energy at $\delta = 0$. **e, f** Differential reflectivity profiles at varying pump-probe delay for $\delta = +5.1$ meV and an average probe intensity of (**e**) 26 mW cm$^{-2}$, below condensation threshold and (**f**) 149 mW cm$^{-2}$, at condensation: measured (red solid lines) and calculated (blue dashed lines). The scale for $\Delta R/R$ is fixed for all plots in (**e**) and (**f**) respectively. The results for the linewidth and energy in a-d were obtained from the fit of the measured profiles to the model.

lived signature is largest, attributed to the buildup time of the condensate PL (Supplementary Fig. 3).

Around threshold, the Stark effect acts both on the uncondensed edges of the spot populated by polaritons and on the condensate being formed at the centre. The former is observed in the shift of the reflectivity dip of the LP, whereas the condensate, manifested in the Stark shift of PL, remains undetected until higher probe intensities, above condensation threshold, yield a strong enough emission. The energy of the undetected condensate grows with probe power[46], such that the condensate will have acquired a sizable blueshift when detected, giving rise to the apparent step-like increase of the energy around threshold. For $\delta \approx 0$, the rate of the interactions inducing the blueshift decreases with the reduced excitonic component of the LP, resulting in a smoother and smaller blueshift.

Notably, the linewidth values presented in Fig. 4 for $\delta \approx 0$ and $\delta = +5.1$ meV are larger than those extracted from angle-resolved PL at $\delta = -2.9$ meV (Supplementary Fig. 5 and Panna et al.[34]). Likewise, spectra extracted from the detuning map of our sample (Supplementary Fig. 6), acquired through reflectivity from a broadband source, also display broader linewidths compared to angle-resolved PL at similar cavity-exciton detuning. These differences arise from the measurement scheme. Since reflectivity measurements capture the full optical response of the system, contributions from all accessible in-plane momenta within the probed range are included in the measured linewidth. In contrast, angle-resolved photoluminescence captures emission from populated states and is further weighted toward efficiently collected, relaxed polaritons near $\mathbf{k}_{\parallel} \approx 0$. As a result, PL measurements typically exhibit a narrower spectral width near $\mathbf{k}_{\parallel} \approx 0$.

### Stark shift at varying pump intensity
Finally, to examine the dependence of the condensate energy shift on the Stark pump power, the differential reflectivity was measured at varying Stark pump peak intensity in the range 2.5–14.7 GW cm$^{-2}$ and for a detuning of $\delta = +5.1$ meV. The shift of the LP and UP energy, depicted by the dark arrows in Fig. 2, is extracted from the measurement of $\Delta R/R$ at small pump-probe time delay $\Delta t \approx 0$ and is shown for a fixed probe (injection) intensity of 13 mW cm$^{-2}$ yielding sub-threshold LP population (Fig. 5a), and for an average probe intensity of 155 mW cm$^{-2}$ yielding condensation (Fig. 5b). The results are compared to the calculated Stark shift as function of pump intensity. The ratio between the extracted Stark shifts of the LP and the UP is consistent with their excitonic fraction (LP is 65% excitonic at $\delta = +5.1$ meV).

Further details concerning the calculation of the Stark shift are provided in Supplementary Note 4.

The Stark shift of the UP is observed below condensation threshold only (Fig. 5a). Its visibility around $\Delta t = 0$ decreases with increasing probe intensity (Fig. 3), likely due to the reflection dip becoming shallower as the branch is more populated. A similar mechanism may also reduce the LP visibility, in combination with linewidth broadening. Above threshold, however, the LP visibility is enhanced by the coherent PL buildup at $\mathbf{k}_{\parallel} \approx 0$, while spectral overlap with the blueshifted LP further contributes to masking the UP feature, which eventually becomes undetectable.

## Discussion
We have demonstrated the reversible and coherent modulation of a nonequilibrium polariton condensate on a femtosecond time scale via the dynamic Stark effect. The Stark-induced shift alongside the formation of the condensate are demonstrated in a time resolved differential reflectance spectroscopy framework. The dynamic Stark effect is featured in an ultrafast energy shift of the polariton branches both below and above the excitation threshold corresponding to condensation of the LP; below threshold, the Stark effect is manifested in the blueshift of the absorption dip of the LP, while above threshold a PL peak is shifted instead.

Further confirmation that the Stark-shifted polaritons are indeed condensed is found in the sharp prolongation of spectro-temporal oscillations at threshold density, signalling the buildup of temporal coherence. Additional evidence is found in the enhanced interaction-induced blueshift of the LP energy typical of condensation. These important characteristics of condensation, reported in extensive PL studies, are here detected for the first time through reflectance spectroscopy

The Stark time-dependent modulation of polariton condensates opens avenues for exploring phenomena of fundamental interest in nonequilibrium solid-state condensates in analogy to existing studies in cold atoms. Our work also paves the way for implementations of polariton-based elements in classical and quantum logics and communication.

## Methods
### Sample details
Our sample, grown by molecular beam epitaxy, features a $\lambda/2$ cavity embedded between two distributed Bragg reflectors (DBR) stacks of 23 and 27 pairs of $Al_{0.2}Ga_{0.8}As/AlAs$ layers at the top and bottom DBRs,

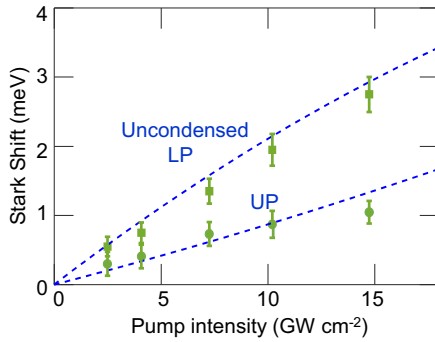

**Fig. 5 | Stark shift of LP and UP versus pump intensity at cavity-exciton detuning $\delta$ = +5.1 meV. a** Measured shift (squares and circles) below condensation threshold, at fixed average probe intensity of 13 mW cm$^{-2}$. **b** Measured shift (squares) above condensation threshold, at fixed average probe intensity of 155 mW cm$^{-2}$. The dashed lines represent the calculated Stark shift. The larger excitonic constituent of the LP, in comparison to the UP, yields a larger Stark shift. The Stark shift of the UP was detected below threshold only. Error bars represent standard deviation.

respectively, resulting in a Q-factor of ~5000. It comprises 3 stacks of 4 GaAs QWs of 7 nm width sandwiched between 4 nm AlAs barriers[34,44] and placed at the antinodes of the resonant field. A wedged cavity structure enables variation of the cavity-exciton detuning by scanning different regions in the sample. The sample incorporates a unique asymmetric design in which the bottom DBR layers are approximately 4% thicker than those of the top DBR, yielding an entrance dip for the 1550 meV Stark pump beam at an angle of 45°, optically compatible with the cryogenic environment. This design supports the use of high-intensity pump beams, as the bottom DBR–substrate interface exhibits high reflectance, effectively limiting substrate heating. A normal mode splitting of 13.6 meV is associated with the strong coupling between the 1s QW HH1 exciton and the cavity mode in the sample[34,44]. The sample was kept at a temperature of 4 K in a liquid-He closed cycle cryostat. The energy of the reflectivity dips of the LP and UP were found to be 1607.4 meV, 1621.5 meV at cavity-exciton detuning $\delta$ = +5.1 meV, respectively, and 1605.3 meV, 1618.9 meV, respectively, at $\delta \approx 0$. The approximate respective spectral widths of the LP and UP are 1.1 meV and 2.1 meV around small cavity-exciton detuning values.

**Differential spectrometry measurement**

The output of a Ti:Sapphire amplifier delivering 1550 meV, 35 fs pulses at a repetition rate of 1 kHz was split into two beams. The first beam pumps an optical parametric amplifier (OPA) tuned to an energy range of 1604 ± 24 meV delivering the probe beam. The direct 1550 ± 31 meV output beam of the amplifier serves as the sub-resonant pump beam inducing the Stark shift. The probe beam was normally incident on the sample and reflected towards a spectrometer while the pump was directed towards the sample at an incidence angle of 45°. One-photon absorption-induced excitation of polaritons by the 1550 meV pump is strongly suppressed as the power of the pump at ~1650 meV, corresponding to the cavity mode within the 45° stopband, is reduced by three orders of magnitude relative to its peak. The pump pulse length is stretched to 250 fs by the dip in reflectivity upon entering the sample. The delay between pump and probe is controlled by a translation stage. The experimental setup is depicted in Supplementary Fig. 7.

It should be noted that the choice of pump-probe ultrafast spectroscopy rather than direct PL measurements stems from the need to capture the femtosecond-scale modifications induced by Stark effect. Since PL signals last several orders of magnitude longer, the impact of the Stark shift on time-integrated PL is negligible. The option of time-resolved measurement of PL with a streak camera would not overcome this limitation, as the temporal resolution of the camera is still limited to the picosecond scale.

The pump and probe beams were linearly polarized along the horizontal direction, and the sample was mounted vertically. Unlike transition metal dichalcogenide monolayers, where valley-selective polariton control using a circularly polarized Stark beam has been reported[47], in our system the rotation direction only distinguishes between spin-degenerate states. It is the orientation of the linear polarization of the oblique pump that has a possible impact as the exciton–light coupling strength may vary between transverse electric (TE) and transverse magnetic (TM) modes[48]. This allows optimization of the Stark effect efficiency by selecting the appropriate polarization. In our case, however, the available laser power exceeded the required power for the Stark effect. Therefore, the pump polarization was not varied, and the magnitude of the Stark shift was determined by the summed contribution of the TE and TM components of the pump.

## Data availability

All data required to evaluate the conclusions are presented in the manuscript and in the Supplementary Materials. The raw data used to generate the figures is available on Zenodo at: https://zenodo.org/records/17877726.

## Code availability

Codes supporting this study are available from the authors upon request.

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

## Acknowledgements

A.H. acknowledges the financial support of the Israel Science Foundation (ISF Grant No. 934/18 and 1605/22). S.H. acknowledges financial support by the German Research Foundation (DFG) within the projects HO 5194/12-1 and HO 5194/20-1.

## Author contributions

A.H. conceived the research idea and supervised the project. The experimental studies were conducted by S.F., with contributions from D.P and N.L. The data analysis was carried out by S.F. with the supervision of A.H. D.P. wrote the simulation. The sample was designed by S.B, S.K., C.S., S.H., A.H., D.P. and N.L. and grown by S.B, S.K., C.S. and S.H. S.F and A.H. wrote the manuscript, with input from all co-authors.

## Competing interests

The authors declare no competing interests.
