## [Transparent Peer Review file · Nature Communications]

Ultrafast Dynamic Stark Shift of an Exciton-Polariton Condensate

Corresponding Author: Ms Sarit Feldman

Version 0:

Reviewer comments:

Reviewer #1

(Remarks to the Author)

In the manuscript, the authors demonstrate the ultrafast, reversible control of polariton condensate via the dynamic Stark effect. The effect is observed through time-resolved differential reflectance spectroscopy. Authors study the dependence of the Stark shift on both pump and probe powers, and observe the shift below and above the condensation threshold, with signatures of condensation confirmed by prolonged coherence and blueshift. Even though the Stark shift was already observed in polariton systems, authors are the first to show it for the polariton condensate. These findings offer a great tool for the manipulation of the polariton condensate on a short time scale that is very useful for the future applications. The manuscript is well-written and structured and presents a concise and interesting study. I will be happy to recommend the manuscript for publication in Nature Communications as long as the Authors address a few comments and concerns arose while I was reading manuscript.

1. Authors claim that the pump is sufficiently red-detuned and does not induce polaritons in the system (line 96); however, later, they refer to the modification of the cavity due to two-photon absorption (line 165). Could this process result in some polariton populating LP and UP and affecting the dynamics at a positive time delay? To clarify, the authors should provide PL spectra with pump-only excitation. Additionally, please include the spectral linewidths of the pump and probe in the experimental setup description.
2. The decay time of oscillations is not characterized quantitatively. The decay time seems to be very close for Figures 2 a and b; knowing the value of the time decay would allow to compare the two probe powers better. Also, Authors claim that the decay time of the oscillations around the LP branch is inversely dependent on density (line 138) due to the increased linewidth of PL. However, the dependence of the linewidth on pump power depicted in Figure 3 a on the opposite demonstrates a decrease of the linewidth for probe intensities of 45 mW/cm², 92 mW/cm² depicted in Figure 2 (a,b), respectively. In this regard, the claim of the inversed dependence of the decay time is now not sufficiently supported by the experimental data.
3. The explanation of the presence of the condensed and uncondensed LP features is not clear. Could the authors elaborate on how the overlap between the collection and excitation paths influences this? Also, could they plot the schematic of the experimental setup in the supplementary materials so this overlap would be clear for the reader?
4. Can the Stark shift be observed in direct PL measurements? What are the advantages of utilizing reflectance spectroscopy here?
5. The manuscript does not clearly explain the physical mechanisms responsible for the smooth versus step-like blueshifts observed at different detunings. A more detailed discussion would be helpful.
6. Why is the decay of the oscillation so long for zero detuning before the threshold (Fig. S1(a))? Authors should comment on how detuning affects the results of the measurements.
7. What was the polarization of the pump and probe? Would the outcome depend on their polarizations?
8. The authors claim the cavity has a unique design; could they add a description of this design to the main text or methods

for clarity?

Small remarks:

1. The notion of bright and opaque purple arrows in Figure 2 (c,d) is mixed throughout the text (lines 136,157).
2. Could authors add a line marking the zero time delay in Figure 2 to improve the comprehensiveness.
3. The (a) and (b) labels are missing from Figure 5
4. μ is not defined for equation (3) in the main text.

Reviewer #2

(Remarks to the Author)

Feldmann et al. report on the energy modulation of a polariton condensate at ultrafast timescales upon excitation by a sub resonant pulse (i.e. without creating additional polaritons), thanks to the dynamic Stark effect. By monitoring the time-resolved pump-probe differential reflectance, for different pump powers, the authors observe a drastic change on its behavior upon crossing a certain power threshold. In particular, the decay of the spectro-temporal oscillations slows down when crossing the threshold, consistent with the formation of a polariton condensate and the onset of temporal coherence associated to it.

The work is certainly original, meaningful and provides a new perspective on polariton condensation, which has been mostly studied up to now by photoluminescence (and coherence) measurements. Thus, this alternative characterization technique and the possibilities it opens in terms of polariton condensate manipulation deserves publication in Nature Communications. However, before it can be accepted for publication a certain number of critical issues need to be clarified, and further analysis and/or additional experiments need to be presented. These are described in detail below. Besides, curing the figures presentation (and figure captions) as well as the text will aid the reader to follow up better the logic behind the results presentation, which is sometimes obscured by the different mechanism in play. In this sense, adding an additional figure (or an additional panel in Figure 2), focusing just on the dynamic Stark shift of the polariton branches at zero delay would be extremely helpful (i.e. not waiting for figure 4 to analyze it in more detail). Otherwise, the interest of this effect is masked by the remarkable oscillations in the differential reflectivity measurements, which are also an important result but not the only one.

A crucial aspect of the study, pushing the current results far beyond those published previously by the authors (ACS Photonics 2019), is related to the transition into a condensed polariton phase and its modulation in a fs time scale. Polariton condensation in the same sample has been reported by the authors and studied thanks to standard techniques, as commonly employed by the polariton community. In their previous results on the same sample, the exciton energy was indicated to be around 1610 meV. However, if I understand correctly Figure 2 (and, Figure 3b and Figure 4b), it seems that the LPB energy at condensation is shifted above the bare exciton energy, which should be clearly stated anyhow. This aspect deserves either a finer analysis or some clear explanation. It might be enough to monitor the transition to the condensed phase with a finer series of pumping power or provide angle-resolved measurements under the same pumping conditions but, in any case, a clear and unambiguous characterization of the polariton branches close to threshold (below and above) is required.

Similarly, the “disappearance” of the UPB signature upon crossing the condensation threshold needs some explanation (and potentially additional measurements). The reasons why the UPB vanishes in reflectivity measurements is not clarified by the authors, and there are no obvious reasons why this should be so (compared to PL measurements). Furthermore, having access to the UPB would allow the reader to compare the blueshift induced by the probe, due to the increase of polariton density, as well as the dynamic Stark shift induced by the Stark pump. In both cases we expect a shift dependent on the excitonic fraction and that, thus, should be different from the shift of the LPB.

If one lets aside the claimed spectral position of the LPB at or above threshold, the LPB linewidth evolution extracted from the time-evolution of the differential reflectance for different probe (i.e. pumping) powers shows clear signatures for the increase of temporal coherence, as expected for the polariton condensation transition. Nonetheless, the linewidth values extracted from the fittings of the differential reflectance spectra seem systematically larger than those extracted previously from angle-resolved PL spectra on the same sample. And this, even if one takes into account the effect of detuning (i.e. of excitonic/photonic fractions), whose values are slightly different in the current and previous works. Again, some clarification is needed to keep consistency between the two measurements and, especially, for the polariton community to understand how to compare (if it is not a direct comparison) between the results obtained by PL measurements and those extracted from the current pump-probe reflectance measurements.

A very interesting aspect of the article is the unequal delay times for the maximum dynamic Stark shift depending on whether the cavity is operated below or above threshold. While the effect is clearly demonstrated from an experimental point of view, its explanation (retarded formation of the condensate) seems somehow contradictory with the fact that the probe beam pumps resonantly the condensate (and the polariton branches below threshold), and thus one would not expect a delay with respect to the formation of the polaritons themselves. It might be due to the fact that, while the probe is resonant, it is still quite broad. Thus, indeed, polaritons contributing to attaining the condensation threshold might need to relax along the LPB from states lying at higher energies. The authors should develop these arguments further, as it is one key observation and deserves some further insight. Furthermore, it would be very useful for the polariton community if the authors provided these unequal delays for a second (or more) detuning, as the effect probably depends on the excitonic/photonic fraction of the state

at which polariton condensation occurs.

Overall, this work provides an insightful view on polariton condensation from an original perspective. The experimental technique can thus provide additional information on the condensation phase transition and, besides, a means for ultrafast polariton condensate control. For all these reasons the referee recommends publication in Nature Communications after the critical issues raised above have been clarified.

Reviewer #3

(Remarks to the Author)

In this manuscript, the authors demonstrate ultrafast, non-invasive modulation of an exciton-polariton condensate using the dynamic Stark effect. By employing a sub-resonant Stark pulse in a femtosecond pump-probe setup, they report transient energy shifts of the lower polariton (LP) condensate, distinguishing between uncondensed and condensed regimes through changes in differential reflectivity spectra. The buildup of coherence in the condensate is inferred from the prolongation of spectro-temporal oscillations, interpreted using a coherent transient model.

While the paper somewhat extends the understanding of nonequilibrium condensation dynamics, several key issues remain that the authors should address:

1. While the authors provide detailed pump-probe data at two cavity-exciton detunings ($\delta = 0$ and $\delta = +5.1$ meV), the manuscript does not present steady-state spectral data (such as reflectivity or PL) under these same detuning conditions. Such a comparison would be useful for identifying how the ultrafast dynamics differ from steady-state behavior, and whether the same trends are observed in the absence of time-resolved perturbation.
2. In Fig. 2, the Stark-induced coherent oscillations appear to abruptly increase in duration between panels (b) and (c), consistent with a condensation threshold. However, one might expect a more gradual shift in the timing of the Stark signal maxima as a function of increasing probe power, especially if the delay is influenced by scattering dynamics. Could the authors clarify why the response appears threshold-like rather than continuous? Additional explanation, or intermediate power data points, may help resolve this point.
3. The authors explain the fitting procedure for Fig. 3 in the Supplementary Information, but similar fitting details are missing for Fig. 5. Specifically, the manuscript does not indicate the form of the fitting function used to model the Stark shift vs. pump intensity, nor are any error bars provided. In Fig. 4, the Stark signal from the condensed LP branch appears to overlap with the UP branch signal. How was this spectral overlap handled in the fitting process? Moreover, the dashed line in Fig. 5, labeled as the “calculated Stark shift,” is not clearly explained in terms of how it was derived—what assumptions or theoretical model were used in this calculation?
4. The title suggests that the condensation process itself is modulated by the dynamic Stark effect, which may be misleading. In the current experiments, the condensate is already formed and is subjected to a transient energy shift induced by the Stark pulse. While this certainly constitutes ultrafast control of the condensate energy, it is not clear that the condensation threshold or coherence properties are themselves being modulated.
5. A key claim of the paper is that the decay time of the coherent oscillations increases significantly upon condensation, reflecting enhanced temporal coherence. However, the manuscript does not provide quantitative data or extracted time constants to support this statement. A more explicit analysis of the decay rates as a function of probe intensity—and, if possible, comparison with theoretical predictions—would help to substantiate this important point.

Version 1:

Reviewer comments:

Reviewer #1

(Remarks to the Author)

The authors have thoroughly addressed all my comments and concerns and have implemented the appropriate changes in the manuscript. I am pleased to recommend it for publication.

Reviewer #2

(Remarks to the Author)

The authors have answered most of the questions I raised during the first revision round, as well as those of the other two referees, and have provided substantial additional and interesting information, which now appears mostly in the Supplementary Information. Besides, the authors have reorganized the sections in the main text, whose logic can now be followed more easily than in the first version.

However, there is still one important question, associated to the blue shifted LPB mode above the exciton energy, that I don't yet understand. The authors refer to two references (Appl. Phys. Lett. 102 (2013) 08115 and Phys. Rev. B 85 (2012) 075318) and claim that polariton condensates blueshifted above the exciton energy were investigated. My understanding of these

two references is that, while there is indeed a large blueshift when the polariton condensate is formed (see for example Figure 3 in the Phys. Rev. B article), its energy never surpasses the energy of the bare exciton energy. This only happens when the second threshold is attained, corresponding to the standard photon lasing. I think this aspect needs to be critically analysed and clearly discussed before the current paper can be published. Otherwise, the reader might understand that some of the experiments could have been conducted across the strong-to-weak coupling regimes transition.

Reviewer #3

(Remarks to the Author)

The authors have well addressed my comments and concerns. The manuscript presents an original and significant demonstration of ultrafast, reversible control of a polariton condensate. I would like to recommend publication of this manuscript.

Version 2:

Reviewer comments:

Reviewer #2

(Remarks to the Author)

The authors have addressed my concern on the spectral position of the LPB and have discussed it appropriately in the manuscript. I am pleased to recommend it for publication

Response to Reviewers

Reviewer #1 (Remarks to the Author):

In the manuscript, the authors demonstrate the ultrafast, reversible control of polariton condensate via the dynamic Stark effect. The effect is observed through time-resolved differential reflectance spectroscopy. Authors study the dependence of the Stark shift on both pump and probe powers, and observe the shift below and above the condensation threshold, with signatures of condensation confirmed by prolonged coherence and blueshift. Even though the Stark shift was already observed in polariton systems, authors are the first to show it for the polariton condensate. These findings offer a great tool for the manipulation of the polariton condensate on a short time scale that is very useful for the future applications. The manuscript is well-written and structured and presents a concise and interesting study. I will be happy to recommend the manuscript for publication in Nature Communications as long as the Authors address a few comments and concerns arose while I was reading manuscript.

We thank the Reviewer for the positive assessment of our work and appreciate the insightful remarks. We have addressed and clarified each of the points raised below.

1. Authors claim that the pump is sufficiently red-detuned and does not induce polaritons in the system (line 96); however, later, they refer to the modification of the cavity due to two-photon absorption (line 165). Could this process result in some polariton populating LP and UP and affecting the dynamics at a positive time delay? To clarify, the authors should provide PL spectra with pump-only excitation. Additionally, please include the spectral linewidths of the pump and probe in the experimental setup description.

Response: Following the Reviewer's remark, the statement was corrected on page 5 (previously in line 96) in order to clarify that the pump detuning prevents excitation by one-photon absorption.

Two-photon absorption (TPA)-induced population of the LP and UP is expected to be negligible; according to our calculations, experimentally validated by pump-only measurements as suggested by the Reviewer, most pump power undergoes TPA in the top

DBR. Further details and the result of pump-only measurement are now included in the supplementary information on page 4, as quoted below.

“Two-photon absorption -induced spectral feature at positive time delays

Two-photon absorption of the pump is expected to be negligible in the QWs. Based on TPA coefficients reported by Nathan et al.[S8] we estimate that while most of the pump power is absorbed in the top distributed Bragg reflector (DBR, only ~0.1% is absorbed by the quantum wells. This estimate is corroborated by photoluminescence (PL) measurements under pump-only excitation in our setup. As shown in Supplementary Figure 1, the PL spectrum exhibits a distinct feature at 1729 meV, close to the bandgap of the Al_{0.2}Ga_{0.8}As DBRs, while no signal is detected at the lower polariton (LP) and upper polariton (UP) LP and UP energies of 1605.3 meV and 1618.9 meV, respectively.

TPA-induced carriers in the DBR account for the deviation of the measured PL at 1729 meV from the nominal cryogenic band gap of Al_{0.2}Ga_{0.8}As (1768 meV); according to Bennett et al.[S9], a carrier density sufficient to induce a bandgap shrinkage of ~30 meV corresponds to a ~15% modification of the DBR refractive index, leading to an expected red shift of 1–2 meV in the polariton energy. This shift is consistent with the feature observed at positive delays in Fig. 3c,d in the main text and Supplementary Figure 4c,d.”

Supplementary Figure 1 | PL spectrum excited with 1550 meV pump. Pump-excited PL was collected from the sample using our experimental setup. A peak corresponding to the bandgap of $\text{Al}_{0.2}\text{Ga}_{0.8}\text{As}$, altered by carrier-induced shrinkage, indicates TPA of the pump in the DBR. No PL is detected for the LP and UP modes at 1605.3 meV and 1618.9 meV, respectively.

Finally, the respective linewidth values of the probe and the pump, of 48 meV and 62 meV, are now mentioned on page 16. The cavity mode at the incidence angle of the pump (45°) is ~ 1650 meV. It is now mentioned in the Methods section on page 16 that since in this spectral region, the pump is reduced by three orders of magnitude relative to its peak at 1550 meV, one-photon absorption–induced excitation of polaritons by the pump is strongly suppressed.

[S8] Nathan, V. et al., Review of multiphoton absorption in crystalline solids. *Journal of the Optical Society of America B* **2**, 294-316 (1985).

[S9] Bennett, B. R. et al., Carrier-induced change in refractive index of InP, GaAs and InGaAsP. *IEEE Journal of Quantum Electronics* **26**, 113-122 (1990).

2. The decay time of oscillations is not characterized quantitatively. The decay time seems to be very close for Figures 2 a and b; knowing the value of the time decay would allow to compare the two probe powers better.

Response: Following the Reviewer’s comment, we addressed the subject in the paper on page 9 and added the values of the decay time in Fig. 3 and Supplementary Figure 4 (previously Fig. 2 and Supplementary Figure 1) along with the given corresponding linewidth values: “The amplitude of the oscillations decays as $e^{-|\Delta t|/\tau}$ where the value of $\tau = 1/\gamma$, indicated for each measurement, is the polariton decay time.”

Fig.3 | Measured normalized differential reflectivity. The probe injects polaritons at an extended range of densities covering condensation threshold. Results in (a)-(d) are shown for an average probe intensity of 45 mW cm^{-2} , 92 mW cm^{-2} , 117 mW cm^{-2} and 149 mW cm^{-2} , respectively at cavity-exciton detuning $\delta = +5.1 \text{ meV}$ and for a fixed pump peak intensity of 10.2 GW cm^{-2} . The LP and the UP are represented by thick purple arrows and green narrow arrows, respectively. The Stark shift at $\Delta t \approx 0$ is associated with the coherent oscillations at $\Delta t < 0$. The onset of condensation, observed in (c), is marked by a significant prolongation of the oscillations. Below condensation threshold (a,b), the oscillatory features are associated with the LP and UP. Above threshold (c,d), the energy of the major part of the LP polaritons is blueshifted, giving rise to the dominant, higher-energy oscillations, while the UP-associated features fade. The bright arrows in (c,d) represent the uncondensed portion of the LP. The dashed lines mark pump-probe temporal overlap ($\Delta t = 0$). τ represents the decay time constant of the oscillations.

Supplementary Figure 4 | Measured normalized differential reflectivity at cavity-exciton detuning $\delta \sim 0$. Results in a-d are shown for an average probe intensity of 16 mW cm^{-2} , 45 mW cm^{-2} , 129 mW cm^{-2} and 162 mW cm^{-2} , respectively and for a pump peak intensity of 10.2 GW cm^{-2} . The LP and the UP are represented by thick purple arrows and green narrow arrows, respectively. The bright thick arrows in (c,d) represent the LPs in the uncondensed state. The dashed lines mark pump-probe temporal overlap, $\Delta t = 0$. τ represents the decay time constant of the oscillations.

Also, Authors claim that the decay time of the oscillations around the LP branch is inversely dependent on density (line 138) due to the increased linewidth of PL. However, the dependence of the linewidth on pump power depicted in Figure 3 a on the opposite demonstrates a decrease of the linewidth for probe intensities of 45 mW/cm^2 , 92 mW/cm^2 depicted in Figure 2 (a,b), respectively. In this regard, the claim of the inversed dependence of the decay time is now not sufficiently supported by the experimental data.

Response: We appreciate the Reviewer's helpful remark. Indeed, a correction was necessary in estimating the linewidth at probe intensities near the condensation threshold. In this regime, as the linewidth broadens, the amplitude of the oscillations decreases and the signal competes with noise at advanced time delays. We therefore shifted the range of delay times employed for fitting the measurements to the model within proper SNR limits for a couple of relevant data points. The revised results now align with the observed inverse relationship between decay time and polariton density below threshold, as illustrated in

Fig. 3 on page 9. We updated the highest values of the linewidth in Fig. 4a accordingly on page 11:

Fig. 4 | Linewidth and energy of the LP branch vs. probe intensity. a,b: Linewidth (a) and energy (b) of the LP branch vs. average probe (injection) intensity at cavity-exciton detuning $\delta = +5.1$ meV. c,d: linewidth and energy at $\delta = 0$. e,f: differential reflectivity profiles at varying pump-probe delay for $\delta = +5.1$ meV and an average probe intensity of (e) 26 mW cm⁻², below condensation threshold and (f) 149 mW cm⁻², at condensation: measured (red solid lines) and calculated (blue dashed lines). The scale for $\Delta R/R$ is fixed for all plots in (e) and (f) respectively. The results for the linewidth and energy in a-d were obtained from the fit of the measured profiles to the model.

3. The explanation of the presence of the condensed and uncondensed LP features is not clear. Could the authors elaborate on how the overlap between the collection and excitation paths influences this? Also, could they plot the schematic of the experimental setup in the supplementary section so this overlap would be clear for the reader?

Response: As suggested by the Reviewer, we rephrased and clarified the explanation for the presence of the condensed and uncondensed features on page 8.

“The coexistence of condensed and uncondensed LP spectral features arises from the nonuniform profile of the spot (the area populated by polaritons) [41]. The polariton density decreases towards the edges of the spot, leading to sub-threshold LPs at the periphery alongside with a condensate at the center. Since the spectrometer collects the signal from a region larger than the spot, the reflectivity spectrum above threshold reflects the contributions of the uncondensed LPs at the edges, in addition to the condensate.”

A sketch of the experimental setup was added to the supplementary information (Supplementary Figure 7):

Supplementary Figure 7 | Experimental sketch of the pump-probe differential spectroscopy measurements setup. The OPA, pumped by the regenerative amplifier, delivers the probe beam, which is focused on the sample by lens L1 and reflected towards the spectrometer. The pump inducing the Stark shift, delivered by the regenerative amplifier, impinges on the sample at oblique incidence and is focused by lens L2. Optical choppers control the alternation between pump and probe.

[41] Deng, Hui, et al. Polariton lasing vs. photon lasing in a semiconductor microcavity. *Proceedings of the National Academy of Sciences* **100**,15318-15323 (2003).

4. Can the Stark shift be observed in direct PL measurements? What are the advantages of utilizing reflectance spectroscopy here?

Response: The choice of reflectance spectroscopy was dictated by the need for high temporal resolution, compatible with the ultrafast dynamics of the Stark effect. Following the Reviewer’s insightful question, we now address the subject in detail in the Methods section on page 17:

“It should be noted that the choice of pump-probe ultrafast spectroscopy rather than direct PL measurements stems from the need to capture the femtosecond-scale modifications induced by Stark effect. Since PL signals last several orders of magnitude longer, the impact of the Stark shift on time-integrated PL is negligible. The option of time-resolved measurement of PL with a streak camera would not overcome this limitation, as the temporal resolution of the camera is still limited to the picosecond scale.”

5. *The manuscript does not clearly explain the physical mechanisms responsible for the smooth versus step-like blueshifts observed at different detunings. A more detailed discussion would be helpful.*

Response: A step-like blueshift is observed at $\delta = +5.1$ meV, where the LP is mainly excitonic, while for $\delta \approx 0$ the shift evolves smoothly. The discontinuity at $\delta = +5.1$ meV originates from the contribution of uncondensed lower polaritons (LP) at the edges of the spot (the area populated by polaritons), detected in reflectivity, and condensed LP at the center, detected in PL. For the condensate to become visible in differential reflectivity, strong PL is required. The latter is achieved at intensities above condensation threshold, for which the blueshift has already grown to large values, resulting in apparent step-like behavior although the energy in fact grows smoothly with excitation power [46]. In contrast, weaker interactions at $\delta \approx 0$ lead to a smooth dependence. We note that although an apparent step-like blueshift in angle-resolved measured PL is also reported in other work [46] and observed in our own sample (Supplementary Figure 2), it is due to temporal integration, rather than to spatial integration as is the case in our differential reflectivity measurements. This topic is now discussed in detail in the revised manuscript on pages 12-13, as quoted below.

“Around threshold, the Stark effect acts both on the uncondensed edges of the spot populated by polaritons and on the condensate being formed at the center. The former is observed in the shift of the reflectivity dip of the LP, whereas the condensate, manifested in the Stark shift of PL, remains undetected until higher probe intensities, above condensation threshold, yield a strong enough emission. The energy of the undetected condensate grows with probe power [46], such that the condensate will have acquired a sizable blueshift when detected, giving rise to the apparent step-like increase of the energy around threshold. For $\delta \approx 0$, the rate of the interactions inducing the blueshift decreases with the reduced excitonic component of the LP, resulting in a smoother and smaller blueshift.”

[46] Tempel, J.S. et al., "Characterization of two-threshold behavior of the emission from a GaAs microcavity", *Physical Review B - Condensed Matter and Materials Physics* **85**, 075318 (2012).

6. *Why is the decay of the oscillation so long for zero detuning before the threshold (Fig. S1(a))? Authors should comment on how detuning affects the results of the measurements.*

Response: The measurement shown in Supplementary Figure 4(a) on page 8 (previously 1a), presenting differential reflectivity at zero detuning, was performed at low excitation intensity, below the range of substantial spectral broadening. Under these conditions, the lower polariton (LP) linewidth is narrower at $\delta \approx 0$ compared to the exciton-like detuning ($\delta = +5.1$ meV), due to the smaller excitonic fraction of the LP. This results in longer coherence times, or equivalently, a slower decay of the oscillations. In response to the Reviewer's remark, a comment discussing the role of detuning has been added to the manuscript on page 12.

“As demonstrated in Fig. 4a,c, when the linewidth is not dominated by thermal broadening - namely at low probe intensities - it is narrower at $\delta \approx 0$ than at $\delta = +5.1$ meV, consistent with the larger photonic fraction of the LP at $\delta \approx 0$. Equivalently, the coherent oscillations decay more slowly.”

7. *What was the polarization of the pump and probe? Would the outcome depend on their polarizations?*

Response: Following the Reviewer's comment, the following clarification was added in the Methods section on pages 17:

“The pump and probe beams were linearly polarized along the horizontal direction, and the sample was mounted vertically. Unlike transition metal dichalcogenide monolayers, where valley-selective polariton control using a circularly polarized Stark beam has been reported [47], in our system the rotation direction of normally incident light only distinguishes between spin-degenerate states. The orientation of the polarization of the oblique pump has a possible impact, as the exciton–light coupling strength may vary between transverse electric (TE) and transverse magnetic (TM) modes [48]. This allows optimization of the Stark effect efficiency by selecting the appropriate polarization . In our

case, however, the available laser power exceeded the required power for the Stark effect. Therefore, the pump polarization was not varied, and the magnitude of the Stark shift was determined by the summed contribution of the TE and TM components of the pump.”

[47] LaMountain, T. et al., Valley-selective optical Stark effect of exciton-polaritons in a monolayer semiconductor. *Nature communications* **12**, 4530 (2021).

[48] Kavokin, A. et al, *Microcavities*, (Oxford University press, 2017).

8. The authors claim the cavity has a unique design; could they add a description of this design to the main text or methods for clarity?

Response: In response to the Reviewer’s suggestion, we have revised the Methods section on pages 15-16 to provide a more detailed description of the sample design, specifically highlighting the modifications that enable stable operation at high pump intensities and accommodate moderate angles of incidence:

“Our sample, grown by molecular beam epitaxy, features a $\lambda/2$ cavity embedded between two distributed Bragg reflectors (DBR) stacks of 23 and 27 pairs of $Al_{0.2}Ga_{0.8}As/AlAs$ layers at the top and bottom DBRs, respectively, resulting in a Q -factor of ~ 5000 . It comprises 3 stacks of 4 GaAs QWs of 7 nm width sandwiched between 4 nm AlAs barriers [34] and placed at the antinodes of the resonant field. A wedged cavity structure enables variation of the exciton-cavity detuning by scanning different regions in the sample. The sample incorporates a unique asymmetric design in which the bottom DBR layers are approximately 4% thicker than those of the top DBR, yielding an entrance dip for the 1550 meV Stark pump beam at an angle of 45° , optically compatible with the cryogenic environment. This design supports the use of high-intensity pump beams, as the bottom DBR–substrate interface exhibits high reflectance, effectively limiting substrate heating. A normal mode splitting of 13.6 meV is associated with the strong coupling between the 1s QW HH1 exciton and the cavity mode in the sample [34],[44]. The sample was kept at a temperature of 4K in a liquid-He closed cycle cryostat. The energy of the reflectivity dips of the LP and UP were found to be 1607.4 meV, 1621.5 meV at cavity-exciton detuning $\delta=+5.1$ meV, respectively, and 1605.3 meV, 1618.9 meV, respectively, at $\delta\approx 0$. The

approximate respective spectral widths of the LP and UP are 1.1 meV and 2.1 meV around small cavity-exciting detuning values.”

[34] Panna, D. et al. Ultrafast manipulation of a strongly coupled light–matter system by a giant AC Stark effect. *ACS Photonics* **6**, 3076-3081 (2019).

[44] Landau, N. et al. Two-photon pumped exciton-polariton condensation. *Optica* **9**, 1347-1352 (2022).

Small remarks:

- 1. The notion of bright and opaque purple arrows in Figure 2 (c,d) is mixed throughout the text (lines 136,157).*
- 2. Could authors add a line marking the zero time delay in Figure 2 to improve the comprehensiveness.*
- 3. The (a) and (b) labels are missing from Figure 5*
- 4. μ is not defined for equation (3) in the main text.*

Response: We thank the Reviewer for pointing these out. The paper was revised accordingly.

Reviewer #2 (Remarks to the Author):

Feldman et al. report on the energy modulation of a polariton condensate at ultrafast timescales upon excitation by a sub resonant pulse (i.e. without creating additional polaritons), thanks to the dynamic Stark effect. By monitoring the time-resolved pump-probe differential reflectance, for different pump powers, the authors observe a drastic change on its behavior upon crossing a certain power threshold. In particular, the decay of the spectro-temporal oscillations slows down when crossing the threshold, consistent with the formation of a polariton condensate and the onset of temporal coherence associated to it.

The work is certainly original, meaningful and provides a new perspective on polariton condensation, which has been mostly studied up to now by photoluminescence (and coherence) measurements. Thus, this alternative characterization technique and the possibilities it opens in terms of polariton condensate manipulation deserves publication in Nature Communications. However, before it can be accepted for publication a certain number of critical issues need to be clarified, and further analysis and/or additional experiments need to be presented. These are described in detail below. Besides, curing the figures presentation (and figure captions) as well as the text will aid the reader to follow up better the logic behind the results presentation, which is sometimes obscured by the different mechanism in play. In this sense, adding an additional figure (or an additional panel in Figure 2), focusing just on the dynamic Stark shift of the polariton branches at zero delay would be extremely helpful (i.e. not waiting for figure 4 to analyze it in more detail). Otherwise, the interest of this effect is masked by the remarkable oscillations in the differential reflectivity measurements, which are also an important result but not the only one.

Response: We thank the Reviewer for the constructive evaluation of our work. We appreciate the positive assessment of our approach and the recognition of its potential impact as well as the identification of points that require clarification and further analysis. We have carefully considered all the comments and have revised the manuscript accordingly.

Following the Reviewer's comment above, we have reordered the sections of the manuscript and now begin by presenting the Stark effect on the uncondensed and condensed polaritons at $\Delta t \approx 0$ in the section titled "Stark effect on LP condensate" which

comprises Fig. 2 (previously Fig. 4) on pages 6-8, prior to the analysis of the coherent oscillations.

A crucial aspect of the study, pushing the current results far beyond those published previously by the authors (ACS Photonics 2019), is related to the transition into a condensed polariton phase and its modulation in a fs time scale. Polariton condensation in the same sample has been reported by the authors and studied thanks to standard techniques, as commonly employed by the polariton community. In their previous results on the same sample, the exciton energy was indicated to be around 1610 meV. However, if I understand correctly Figure 2 (and, Figure 3b and Figure 4b), it seems that the LPB energy at condensation is shifted above the bare exciton energy, which should be clearly stated anyhow. This aspect deserves either a finer analysis or some clear explanation. It might be enough to monitor the transition to the condensed phase with a finer series of pumping power or provide angle-resolved measurements under the same pumping conditions but, in any case, a clear and unambiguous characterization of the polariton branches close to threshold (below and above) is required.

Response: Indeed, above the condensation threshold, the LPB energy exceeds the bare exciton energy of approximately 1612 meV. Such a pronounced blueshift has been reported in previous studies of condensates, particularly at positive detuning [45]. Angle-resolved PL measurements under off-resonant excitation performed on our sample also show LP energies reaching the bare exciton energy above threshold at $\delta \approx 0$, as shown in Supplementary Note 3 (presented below). These results are comparable to the case of $\delta > 0$ observed in our reflectivity setup. In addition, the following comment regarding the trend in the LP energy in the measured differential reflectivity is now presented in the manuscript on page 12.

“Pronounced blueshifts, similar to our results at $\delta = +5.1$ meV where the LP energy can exceed the bare exciton energy, are observed in previously reported PL measurements at

positive up to zero detuning in other work [45] as well as in our own sample (Supplementary Note 3).

“Supplementary Note 3: LP energy versus excitation power in PL measurements

The LP energy versus excitation power obtained in PL measurements of our sample at $\delta \approx 0$ is shown in Supplementary Figure 2. The results were obtained using a resonant pump incident on the sample at 45° , tuned to 1653 meV to match the cavity dip associated with the 45° stopband, and the PL was collected using an objective lens. We note that the discontinuity in the blueshift observed here as well as in PL measurements reported for other samples [S10] is an apparent effect due to pulsed excitation; in fact, the energy of the LP varies smoothly with excitation power. This effect results from temporal integration of the measured PL, whereas in our differential reflectivity measurements the nature of the effect is spatial, as described in the main text.”

Supplementary Figure 2 | LP energy versus average excitation power in direct PL measurements. The results are shown for cavity-exciton detuning $\delta \approx 0$. The excitation beam, impinging on the sample at oblique incidence, was centered at 1653 meV, corresponding to the cavity dip within the 45° stopband. The resulting PL was collected using an objective lens. A sharp blueshift is observed, as is the case in our experiment for $\delta = 5.1$ meV.

[45] Schmutzler, J. et al. Determination of operating parameters for a GaAs-based polariton laser, *Applied Physics Letters* **102** (2013).

[S10] Tempel, J.S. et al., "Characterization of two-threshold behavior of the emission from a GaAs microcavity", *Physical Review B - Condensed Matter and Materials Physics* **85**, 075318 (2012).

Similarly, the “disappearance” of the UPB signature upon crossing the condensation threshold needs some explanation (and potentially additional measurements). The reasons why the UPB vanishes in reflectivity measurements is not clarified by the authors, and there are no obvious reasons why this should be so (compared to PL measurements). Furthermore, having access to the UPB would allow the reader to compare the blueshift induced by the probe, due to the increase of polariton density, as well as the dynamic Stark shift induced by the Stark pump. In both cases we expect a shift dependent on the excitonic fraction and that, thus, should be different from the shift of the LPB.

Response: In response to the Reviewer’s comment, we have added further clarification regarding the UP to the manuscript on page 14:

“The Stark shift of the UP is observed below condensation threshold only (Fig. 5a). Its visibility around $\Delta t = 0$ decreases with increasing probe intensity (Fig.3), likely due to the reflection dip becoming shallower as the branch is more populated. A similar mechanism may also reduce the LP visibility, in combination with thermal linewidth broadening. Above threshold, however, the LP visibility is enhanced by the coherent PL buildup at $k \approx 0$, while spectral overlap with the blueshifted LP further contributes to masking the UP feature, which eventually becomes undetectable.”

As a result, the UP energy shift induced by the Stark pump can only be reliably tracked below the LP condensation threshold. The dependence of this shift on pump intensity, calculated for the uncondensed UP state at detuning $\delta = +5$ meV, is now included in Fig.5 alongside the LP shift. The UP energy extracted below condensation exhibits no dependence on probe intensity and remains fixed around 1621.5 meV, as is now mentioned in the paper on page 12.

Fig. 5 | Stark shift of LP and UP versus pump intensity at cavity-exciton detuning $\delta = 5.1$ meV. **a.** Measured shift (squares and circles) below condensation threshold, at fixed average probe intensity of 13 mW cm^{-2} . **b.** Measured shift (squares) above condensation threshold, at fixed average probe intensity of 155 mW cm^{-2} . The dashed lines represent the calculated Stark shift. The larger excitonic constituent of the LP, in comparison to the UP, yields a larger Stark shift. The Stark shift of the UP was detected below threshold only. Error bars represent standard deviation.

If one lets aside the claimed spectral position of the LPB at or above threshold, the LPB linewidth evolution extracted from the time-evolution of the differential reflectance for different probe (i.e. pumping) powers shows clear signatures for the increase of temporal coherence, as expected for the polariton condensation transition. Nonetheless, the linewidth values extracted from the fittings of the differential reflectance spectra seem systematically larger than those extracted previously from angle-resolved PL spectra on the same sample. And this, even if one takes into account the effect of detuning (i.e. of excitonic/photonic fractions), whose values are slightly different in the current and previous works. Again, some clarification is needed to keep consistency between the two measurements and, especially, for the polariton community to understand how to compare (if it is not a direct comparison) between the results obtained by PL measurements and those extracted from the current pump-probe reflectance measurements.

Response: We thank the Reviewer for raising this valuable point which we now address in the paper on page 13 as quoted below. Profiles of the sample detuning map displaying reflectivity spectra at varying cavity-exciton detuning values, and the linewidth measured

in PL measurements, were added in the supplementary information (Supplementary Figure 6 and Supplementary Figure 5, shown below).

“Notably, the linewidth values presented in Fig. 4 for $\delta \approx 0$ and $\delta = +5.1$ meV are larger than those extracted from angle-resolved PL at $\delta = -2.9$ meV (Supplementary Fig. 5, ref. 34). Likewise, spectra extracted from the detuning map of our sample (Supplementary Figure 6), acquired through reflectivity from a broadband source, also display broader linewidths compared to angle-resolved PL at similar cavity-exciton detuning. These differences arise from the measurement scheme. Since reflectivity measurements capture the full optical response of the system, contributions from all accessible in-plane momenta within the probed range are included in the measured linewidth. In contrast, angle-resolved photoluminescence captures emission from populated states and is further weighted toward efficiently collected, relaxed polaritons near $k_{\parallel} \approx 0$. As a result, PL measurements typically exhibit a narrower spectral width near $k_{\parallel} \approx 0$.”

Supplementary Figure 6 | Normalized reflectivity spectra at varying detuning. The profiles display the measured reflectivity spectra acquired at $T=4\text{K}$ for different exciton-cavity detuning values with a broadband light source. The labels δ , ΔE_{LP} and ΔE_{UP} denote the respective values of the detuning, the extracted linewidth of the LP and the extracted linewidth of the UP.

Supplementary Figure 5 | Energy and linewidth of PL versus average excitation power. The energy (a) and linewidth (b) were extracted from angle-resolved PL spectral measurements of our sample at cavity-exciton detuning $\delta = -2.9$ meV with off-resonant pulsed excitation. The onset of condensation is accompanied by a discontinuity in the blueshift and linewidth narrowing.

A very interesting aspect of the article is the unequal delay times for the maximum dynamic Stark shift depending on whether the cavity is operated below or above threshold. While the effect is clearly demonstrated from an experimental point of view, its explanation (retarded formation of the condensate) seems somehow contradictory with the fact that the probe beam pumps resonantly the condensate (and the polariton branches below threshold), and thus one would not expect a delay with respect to the formation of the polaritons themselves. It might be due to the fact that, while the probe is resonant, it is still quite broad. Thus, indeed, polaritons contributing to attaining the condensation threshold might need to relax along the LPB from states lying at higher energies. The authors should develop these arguments further, as it is one key observation and deserves some further insight. Furthermore, it would be very useful for the polariton community if the authors provided these unequal delays for a second (or more) detuning, as the effect probably depends on the excitonic/photon fraction of the state at which polariton condensation occurs.

Response: We agree with the Reviewer's observation that the broad probe pump states which subsequently relax along the lower polariton branch (LPB). This relaxation process

accounts for the delay observed in the timing of the maximal Stark shift in the condensed regime. Below condensation threshold, the Stark effect appears as an energy shift of the reflectivity dips, hence its timing corresponds to the buildup of the system polarization. However, above threshold, the condensate PL dominates, and the Stark effect manifests as a shift in the PL. Consequently, the maximal effect is delayed by relaxation into the $k_{\parallel} \approx 0$ states associated with the observed PL.

Comparison to zero detuning ($\delta \approx 0$), as suggested by the Reviewer, shows that above threshold excitation intensities, the delay times are slightly shorter for exciton-like detuning of the LP ($\delta = +5.1$ meV) than for $\delta \approx 0$. This finding supports the interpretation that relaxation dynamics governs the delay in the condensate. The reduced excitonic fraction and possibly the deeper dispersion curve at $\delta \approx 0$ result in lower scattering rates compared to $\delta = +5.1$ meV. We have clarified the explanation of the delay and included the comparison between zero and positive detuning in the manuscript on page 7, as well as provided the following detailed delay time data for all measurements in the supplementary information:

“Below condensation threshold, the Stark effect appears as an energy shift of the reflectivity dips, with its timing corresponding to the buildup of the system polarization. Above the threshold, the Stark shift manifested in the condensate PL reaches its maximum once the condensate has formed. The delay of this shift corresponds to the time required for relaxation into the $k_{\parallel} \approx 0$ states from higher energy states excited by the resonant, but broadband, probe beam. This contrast in delay times between the condensed and uncondensed systems persists across all probe intensities for $\delta = +5.1$ meV and $\delta \approx 0$ (Supplementary Figure 3). The delay times for the condensate are slightly shorter for the exciton-like detuning of $\delta = +5.1$ meV compared to $\delta \approx 0$, consistent with the longer relaxation times expected for an increased excitonic fraction and a shallower dispersion curve.”

Supplementary Figure 3 | Pump-probe delay at which the Stark effect on the LP is maximal (a) at cavity-exciton detuning $\delta=+5.1$ meV (b) at cavity-exciton detuning $\delta\approx 0$. The delay increases above condensation threshold, when the effect is manifested in PL from the condensate. The delay times above condensation threshold are slightly shorter for the exciton-like detuning of $\delta = +5.1$ meV than for $\delta \approx 0$.

Overall, this work provides an insightful view on polariton condensation from an original perspective. The experimental technique can thus provide additional information on the condensation phase transition and, besides, a means for ultrafast polariton condensate control. For all these reasons the referee recommends publication in Nature Communications after the critical issues raised above have been clarified.

Reviewer #3 (Remarks to the Author):

In this manuscript, the authors demonstrate ultrafast, non-invasive modulation of an exciton-polariton condensate using the dynamic Stark effect. By employing a sub-resonant Stark pulse in a femtosecond pump-probe setup, they report transient energy shifts of the lower polariton (LP) condensate, distinguishing between uncondensed and condensed regimes through changes in differential reflectivity spectra. The buildup of coherence in the condensate is inferred from the prolongation of spectro-temporal oscillations, interpreted using a coherent transient model. While the paper somewhat extends the understanding of nonequilibrium condensation dynamics, several key issues remain that the authors should address:

1. While the authors provide detailed pump-probe data at two cavity-exciton detunings ($\delta = 0$ and $\delta = +5.1$ meV), the manuscript does not present steady-state spectral data (such

as reflectivity or PL) under these same detuning conditions. Such a comparison would be useful for identifying how the ultrafast dynamics differ from steady-state behavior, and whether the same trends are observed in the absence of time-resolved perturbation.

Response: We thank the Reviewer for his constructive comments and for the insightful review of our manuscript. We have carefully addressed the key issues raised and revised the manuscript accordingly.

In accordance to the Reviewer's suggestion, we have added reflectivity and PL measurements to the supplementary information (Supplementary Figure 6 and Supplementary Figure 5), as detailed below. Supplementary Figure 6 presents normalized reflectivity profiles acquired using a broadband CW light source at various detuning values, including $\delta \sim 0$ and $\delta \sim +5$ meV. PL measurements, demonstrated in Supplementary Figure 5, were obtained by off-resonant pulsed excitation at the energy corresponding to the first dip above the stop band at varying excitation power below and above condensation threshold for $\delta = -2.9$ meV.

Supplementary Figure 6 | Normalized reflectivity spectra at varying detuning. The profiles display the measured reflectivity spectra acquired at $T=4\text{K}$ for different exciton-cavity detuning values with a broadband light source. The labels δ , ΔE_{LP} and ΔE_{UP} denote the respective values of the detuning, the extracted linewidth of the LP and the extracted linewidth of the UP.

Supplementary Figure 5 | Energy and linewidth of PL versus average excitation power. The energy (a) and linewidth (b) were extracted from angle-resolved PL spectral measurements of our sample at cavity-exciton detuning $\delta=-2.9\text{ meV}$ with off-resonant pulsed excitation. The onset of condensation is accompanied by a discontinuity in the blueshift and linewidth narrowing.

2. In Fig. 2, the Stark-induced coherent oscillations appear to abruptly increase in duration between panels (b) and (c), consistent with a condensation threshold. However, one might expect a more gradual shift in the timing of the Stark signal maxima as a function of increasing probe power, especially if the delay is influenced by scattering dynamics. Could the authors clarify why the response appears threshold-like rather than continuous? Additional explanation, or intermediate power data points, may help resolve this point.

Response: The timing of the Stark maxima versus excitation intensity, extracted for all intensity values plotted in Fig. 4a-d (previously Fig. 3), indeed exhibits threshold-like behavior at excitonic LP detuning ($\delta=+5.1$ meV). Following the Reviewer's suggestion, the results for cavity-exciton detuning $\delta\sim 0$ and $\delta=+5.1$ meV are now presented in Supplementary Figure 3, as shown below.

The threshold excitation intensity for the time delay roughly corresponds not just to the linewidth threshold, but also to the transition of the Stark shift signature from a blueshifted reflectivity dip to a blueshifted PL peak. This transition, demonstrated for example in Fig. 2, is responsible for the abrupt rise in the timing of the Stark signal maxima and can be explained as follows: below condensation threshold, the Stark effect appears as an energy shift of the reflectivity dips, hence its timing corresponds to the buildup of the system polarization. However, above threshold, the condensate PL dominates, and the Stark effect manifests as a shift in the PL rather than the reflectivity dips. Consequently, the maximal effect is delayed by relaxation into the $k_{\parallel}\approx 0$ states (from states excited at higher k_{\parallel} by the broad probe) associated with the observed PL. This point was clarified in the paper on page 8.

Supplementary Figure 3 | Pump-probe delay at which the Stark effect on the LP is maximal (a) at cavity-exciton detuning $\delta = +5.1$ meV (b) at cavity-exciton detuning $\delta \approx 0$. The delay increases above condensation threshold, when the effect is manifested in PL from the condensate. The delay times above condensation threshold are slightly shorter for the exciton-like detuning of $\delta = +5.1$ meV than for $\delta \approx 0$.

3. The authors explain the fitting procedure for Fig. 3 in the supplementary section, but similar fitting details are missing for Fig. 5. Specifically, the manuscript does not indicate the form of the fitting function used to model the Stark shift vs. pump intensity, nor are any error bars provided. In Fig. 4, the Stark signal from the condensed LP branch appears to overlap with the UP branch signal. How was this spectral overlap handled in the fitting process? Moreover, the dashed line in Fig. 5, labeled as the “calculated Stark shift,” is not clearly explained in terms of how it was derived—what assumptions or theoretical model were used in this calculation?

Response: In response to the Reviewer’s comment, the fitting procedure and theoretical model for the Stark shift are now described in the Supplementary Note 4 on page 7:

“To estimate the Stark shift as a function of pump intensity, a reference spectral profile, corresponding to the LP or UP reflectivity dip without the Stark beam was subtracted from

a profile shifted by varying Stark shifts, ΔE_{Stark} . The estimated shift ΔE_{Stark} was found according to the result which best fitted the measured differential reflectivity.”

The theoretical Stark shift, shown as dashed lines in Fig. 5 on page 14, is obtained from a Hamiltonian describing the Stark-induced modification of the polariton energies, as now detailed in the supplementary section:

“The shift was derived from the eigenstates of the following Hamiltonian representing the system of an exciton interacting with the microcavity vacuum field and with the pump, with the pump coupling efficiency as a fitting parameter to the calculated results:

$$H = \begin{pmatrix} E_x & \hbar\Omega_R & \hbar\Omega_p \\ \hbar\Omega_R & E_c & 0 \\ \hbar\Omega_p & 0 & E_p \end{pmatrix} \quad (1)$$

E_x , E_c and E_p are the exciton energy, cavity mode and energy of the Stark beam, respectively. Ω_R represents the dipole interaction strength of the exciton with the cavity photon, and Ω_p is the interaction strength with the Stark field. The pump-exciton coupling coefficient was set as a free parameter. The calculated theoretical Stark shift is given by the difference between the eigenvalues of H , $E_{LP,S}$ and $E_{UP,S}$ and the energy of the unperturbed LP and UP energy, E_{LP} and E_{UP} :

$$\begin{aligned} \Delta E_{LP} &= E_{LP,S} - E_{LP} \\ \Delta E_{UP} &= E_{UP,S} - E_{UP} \end{aligned} \quad (2)$$

The fitting procedure for the Stark effect in condensed LP does not include the effect of the overlap with the UP branch since its signature is reduced at high probe intensity. The following clarification was added on page 14:

“The Stark shift of the UP is observed below condensation threshold only (Fig. 5a). Its visibility around $\Delta t = 0$ decreases with increasing probe intensity (Fig.3), likely due to the reflection dip becoming shallower as the branch is more populated. A similar mechanism may also reduce the LP visibility, in combination with thermal linewidth broadening. Above

threshold, however, the LP visibility is enhanced by the coherent PL buildup at $k \approx 0$, while spectral overlap with the blueshifted LP further contributes to masking the UP feature, which eventually becomes undetectable.”

The Stark shift dependence on pump intensity of the UP below condensation threshold is now included in Fig. 5 alongside the Stark shift of the LP. Additionally, error bars have been included.

Fig. 5 | Stark shift of LP and UP versus pump intensity at cavity-exciton detuning $\delta = 5.1$ meV. **a.** Measured shift (squares and circles) at occupation below condensation threshold, at fixed average probe intensity of 13 mW cm^{-2} **b.** Measured shift (squares) above condensation threshold, at fixed average probe intensity of 155 mW cm^{-2} . The dashed lines represent the calculated Stark shift. The larger excitonic constituent of the LP, in comparison to the UP, yields a larger Stark shift. The Stark shift of the UP was detected below threshold only. Error bars represent one standard deviation.

4. The title suggests that the condensation process itself is modulated by the dynamic Stark effect, which may be misleading. In the current experiments, the condensate is already formed and is subjected to a transient energy shift induced by the Stark pulse. While this certainly constitutes ultrafast control of the condensate energy, it is not clear that the condensation threshold or coherence properties are themselves being modulated.

Response: In response to the Reviewer’s remark, the title was changed to: “Ultrafast Dynamic Stark Shift of an Exciton-Polariton Condensate”.

5. A key claim of the paper is that the decay time of the coherent oscillations increases significantly upon condensation, reflecting enhanced temporal coherence. However, the manuscript does not provide quantitative data or extracted time constants to support this

statement. A more explicit analysis of the decay rates as a function of probe intensity—and, if possible, comparison with theoretical predictions—would help to substantiate this important point.

Response: Following the Reviewer’s comment, we addressed the subject in the paper and added the values of the decay time in Fig. 3 (previously Fig. 2) on page 9 and Supplementary Figure 4 on page 8 along with the given corresponding linewidth values.

“The amplitude of the oscillations decays as $e^{-|\Delta t|/\tau}$ where the value of $\tau = 1/\gamma$, indicated for each measurement, is the polariton decay time.”

Fig. 3 | Measured normalized differential reflectivity at cavity-exciton detuning $\delta \sim 0$. Results in a-d are shown for an average probe intensity of 16 mW cm^{-2} , 45 mW cm^{-2} , 129 mW cm^{-2} and 162 mW cm^{-2} , respectively and for a pump peak intensity of 10.2 GW cm^{-2} . The LP and the UP are represented by thick purple arrows and green narrow arrows, respectively. The bright thick arrows in (c,d) represent the LPs in the uncondensed state. The dashed lines mark pump-probe temporal overlap, $\Delta t=0$. τ represents the decay time constant of the oscillations.

Supplementary Figure 4 | Measured normalized differential reflectivity at cavity-exciton detuning $\delta \sim 0$. Results in a-d are shown for an average probe intensity of 16 mW cm^{-2} , 45 mW cm^{-2} , 129 mW cm^{-2} and 162 mW cm^{-2} , respectively and for a pump peak intensity of 10.2 GW cm^{-2} . The LP and the UP are represented by thick purple arrows and green narrow arrows, respectively. The bright thick arrows in (c,d) represent the LPs in the uncondensed state. The dashed lines mark pump-probe temporal overlap, $\Delta t = 0$. τ represents the decay time constant of the oscillations.

Response to Reviewers

Reviewer #1 (Remarks to the Author):

The authors have thoroughly addressed all my comments and concerns and have implemented the appropriate changes in the manuscript. I am pleased to recommend it for publication.

Response: We thank the Reviewer for the recommendation and for the contribution of their insightful comments.

Reviewer #2 (Remarks to the Author):

The authors have answered most of the questions I raised during the first revision round, as well as those of the other two referees, and have provided substantial additional and interesting information, which now appears mostly in the Supplementary Information. Besides, the authors have reorganized the sections in the main text, whose logic can now be followed more easily than in the first version.

However, there is still one important question, associated to the blue shifted LPB mode above the exciton energy, that I don't yet understand. The authors refer to two references (Appl. Phys. Lett. 102 (2013) 08115 and Phys. Rev. B 85 (2012) 075318) and claim that polariton condensates blueshifted above the exciton energy were investigated. My understanding of these two references is that, while there is indeed a large blueshift when the polariton condensate is formed (see for example Figure 3 in the Phys. Rev. B article), its energy never surpasses the energy of the bare exciton energy. This only happens when the second threshold is attained, corresponding to the standard photon lasing. I think this aspect needs to be critically analysed and clearly discussed before the current paper can be published. Otherwise, the reader might understand that some of the experiments could have been conducted across the strong-to-weak coupling regimes transition.

Response: We thank the Reviewer for his valuable comment. Indeed, at zero cavity-exciton detuning where the exciton bare energy coincides with the cavity energy, the energy of the blueshifted lower polariton shown in [46] and [45], does not surpass the exciton energy in the polariton condensate regime. With increasing positive detuning and enhanced excitonic

fraction, larger blueshifted energy values are reported, eventually exceeding the bare exciton beyond a certain positive detuning value (Fig. 3 in [45]). In our measurements, two additional observations confirm that the blueshift corresponds to a polariton condensate. The first is the short-lived signature of the Stark shift feature visible in Fig. 2b (shown below), indicating the presence of an excitonic component interacting with the Stark beam. In addition, a prolongation of the pump-probe delay Δt for which the Stark effect is maximal is manifested above threshold (dashed lines in Fig. 2b versus Fig. 2a). This prolongation is attributed to the fact that above threshold, where the Stark beam induces a shift in a strong PL rather than absorption dips, this delay is determined by the buildup time of the condensate. Following the Reviewer's comment, the subject has been clarified on page 12:

“The high values of the blueshifted LP energy at cavity–exciton detuning $\delta = +5.1$ meV, compared to the energy values obtained at $\delta \approx 0$, characterize polariton condensates formed at positive detuning [45], where the excitonic fraction of the LP is enhanced. Measurements of the LP energy versus excitation intensity at $\delta \approx 0$ obtained using a PL setup, are provided for reference in Supplementary Note 3. To confirm that the blueshift originates from condensed polaritons and to rule out attributing it to photon lasing, we point out both to the short-lived signature of the Stark shift around $\Delta t \approx 0$, well above the bare exciton energy (Fig. 2b, Fig. 3c,d), and to the relaxation-induced prolongation above condensation threshold of the pump-probe delay at which the short-lived signature is largest, attributed to the buildup time of the condensate PL (Supplementary Figure 3).”

Fig. 1 | Normalized differential reflectivity measured at high temporal resolution. The measurements shown in (a),(b) were acquired around $\Delta t=0$ at detuning $\delta=+5.1$ meV and at average probe intensity of 65 mW cm^{-2} and 130 mW cm^{-2} , respectively, corresponding to a density below and above condensation threshold. a. Stark shift of the unpopulated branches of the LP (thick arrows) and the UP (thin arrow) branches. b. Stark effect acting on the co-existing LP states consisting of the uncondensed portion of the LP (thick bright arrow) and of the blueshifted polariton condensate, manifested as a shift of the PL energy (thick dark arrow). The dashed lines mark the pump-probe delay at which the effect is maximal.

[46] Tempel, J.S. et al., "Characterization of two-threshold behavior of the emission from a GaAs microcavity", *Physical Review B - Condensed Matter and Materials Physics* **85**, 075318 (2012).

[45] Schmutzler, J. et al. Determination of operating parameters for a GaAs-based polariton laser, *Applied Physics Letters* **102** (2013).

Reviewer #3 (Remarks to the Author):

The authors have well addressed my comments and concerns. The manuscript presents an original and significant demonstration of ultrafast, reversible control of a polariton condensate. I would like to recommend publication of this manuscript

Response: We greatly appreciate the Reviewer's positive evaluation, helpful comments and the recommendation of our manuscript for publication.